# Prevalence and Economic Costs of Absenteeism in an Aging Population—A Quasi-Stochastic Projection for Germany

Patrizio Vanella [1,2,*] , Christina Benita Wilke [3] and Doris Söhnlein [4]

1 Helmholtz Centre for Infection Research, Epidemiology Department, 38124 Brunswick, Germany
2 Chair of Empirical Methods in Social Science and Demography, Faculty of Economics and Sociology, University of Rostock, 18057 Rostock, Germany
3 Chair of Economics, Faculty of Economics, FOM University of Applied Sciences, Hochschulzentrum Bremen, 28359 Bremen, Germany; christina.wilke@fom.de
4 Institute for Employment Research (IAB), Forecasts and Macroeconomic Analyses Department, 90478 Nürnberg, Germany; doris.soehnlein@iab.de
* Correspondence: patrizio.vanella@helmholtz-hzi.de

**Abstract:** Demographic change is leading to the aging of German society. As long as the baby boom cohorts are still of working age, the working population will also age—and decline as soon as this baby boom generation gradually reaches retirement age. At the same time, there has been a trend toward increasing absenteeism (times of inability to work) in companies since the zero years, with the number of days of absence increasing with age. We present a novel stochastic forecast approach that combines population forecasting with forecasts of labor force participation trends, considering epidemiological aspects. For this, we combine a stochastic Monte Carlo-based cohort-component forecast of the population with projections of labor force participation rates and morbidity rates. This article examines the purely demographic effect on the economic costs associated with such absenteeism due to the inability to work. Under expected future employment patterns and constant morbidity patterns, absenteeism is expected to be close to 5 percent by 2050 relative to 2020, associated with increasing economic costs of almost 3 percent. Our results illustrate how strongly the pronounced baby boom/baby bust phenomenon determines demographic development in Germany in the midterm.

**Keywords:** cohort-component method; multivariate methods; time series analysis; Monte Carlo methods; stochastic forecasting; demography; statistical epidemiology; labor market research; health economics

## 1. Introduction

A demographic transition is leading to the aging of industrialized societies. The median age in the European Union (EU), for instance, has increased from about 34 to 43 years between 1985 and 2019 [1]. Those trends also hold for other economically developed regions of the world. In East Asia, for instance, aging patterns are even more distinct. Japan, which serves as a good case study, has seen a steady increase in the share of its population aged 65 and above after World War II, starting at a value below 5% in the 1940s and reaching a value of 29% in 2020 [2]. This has strong implications for the labor market, as this trend is associated with an aging workforce [3]. The mentioned regions saw sharp increases in fertility during the two decades following World War II, often referred to as a baby boom. Those born during that phase are therefore often labeled the baby boomers [4].

In the case of Germany, the baby boomers are currently in their last years of labor force participation. The 1964 cohort, which has seen the highest postwar birth number [5], turned 55 in 2019 and is expected to regularly retire at age 67 in 2031 [6]. In the short term, therefore, the German economy will need to focus on the aging of the workforce while in the medium and long term, the shrinking and rejuvenation process will prevail [7]. One important issue to address, among others, is how this demographic development will affect

the future economic costs of companies and the German health insurance system with respect to absenteeism.

Absenteeism describes periods of health-related incapacity to work. Early studies on work absence investigated the link between demographics and work absence due to various causes or to a specific illness for the United States (see, e.g., [8]). Large strands of the literature since then have investigated absenteeism from a rather microeconomic perspective, i.e., applying cost-utility frameworks that interpret work absence as an employee's decision whether or not to attend work. These approaches take the potential costs of absence into account, including the distribution of absence costs, which may depend on the institutional context, i.e., the country or whether or not there are unions in the respective field of work, for instance [9]. While this perspective is rather cross-sectional, we are specifically interested in longitudinal trends in absenteeism in forecasting, which may serve as valuable information for political and economic planning [10]. For instance, effective monitoring of teaching to ensure the learning process among students could include investigations of long-term trends in teachers' absenteeism [11]. Another example is the scheduling of hospital staff on duty to ensure proper care for the patients [12]. These are just two practical examples of cases in which forecasts of absenteeism are of major importance.

Although aging is a well-researched topic in the contexts of labor supply [3], pensions [6], healthcare (costs) [4], or long-term care and disabilities [13], the question of absenteeism going along with aging has been given much less attention. It is, after all, not only a question of how strong the labor supply will be affected by aging but also to what extent it is threatened to be temporarily lost due to (chronic) diseases and illnesses. The present aging report by the European Union (EU), for instance, does not even mention the problem of absenteeism [4]. Trends in increasing absenteeism, however, are becoming a major concern. In the old EU-27 countries, for instance, the sum of quarterly absences associated with illness or disability increased from quarter 4, 2006 to quarter 4, 2019 from 3 million to 4.2 million [14]. Although these developments stress the importance of more thorough investigations of longitudinal trends in absenteeism, i.e., in forecast studies, previous studies addressing absenteeism have remained rather descriptive, not offering projections of absenteeism in the context of the demographic transition—although it is well-known that absenteeism occurs more often and for a longer period later in work life [15].

In Germany, which is the most populous country in the EU [4], aging is a topic of major concern given the country's low fertility [16] and decreasing mortality [17] for almost half a century. At the same time, since the mid-2000s, a trend toward increasing absenteeism can be observed, as the average number of days of absenteeism per case increases with age [18]. Is this trend likely to continue in the future? What economic costs are already incurred by companies because of absenteeism, and what costs can ceteris paribus be expected in the future as a result of the German demographic development? These are the questions we address in this paper.

How the working-age population will develop in the future depends on the underlying population trends. These are determined by the three major demographic components: fertility, migration, and mortality [7]. Of particular relevance for the development of the labor force are the fertility rates, which have a delayed effect on the labor force when the newborn enter working age [16], and migration, which affects the labor market relatively quickly as people tend to migrate when they are of working age. However, to what extent migrants also succeed in entering the German labor market, very much depends on the migrants' sociodemographic background [3,19]. For instance, the labor market integration of refugees takes significantly more time on average than for foreign citizens from the EU [19]. Mortality trends, on the contrary, have a rather negligible impact on the labor force, as they become quantitatively noticeable at older ages, long after the end of the common working life [3]. In this paper, we will project future labor force development driven by these components. Apart from demographic trends, the development of absenteeism is affected by employment trends, which are covered well by labor force participation rates

(LFPRs) (see, e.g., [3]) and epidemiological developments, as represented by morbidity, incidence, or prevalence rates (see, e.g., [8,18]). A holistic forecast of absenteeism should cover this multitude of aspects.

Previous studies suggest forecast approaches for absenteeism, falling short, however, of elaborating long-term forecasts in the end. For instance, Lima et al. [20] compared the performance of four different machine learning approaches in predicting absenteeism among security agents in Brazil over a six-year period, falling short, however, of presenting a long-term forecast. Moreover, they did not include labor market trends in their model. Furthermore, these approaches require microlevel data, which often are not available to forecasters. Msosa [11] suggested classical time series approaches, especially the Holt–Winters model, for forecasting regional absenteeism rates among teachers in South Africa, yet presented neither a real forecast nor included demographic, economic, or epidemiological trends in the model. Markham and Markham [21] presented a prediction model of absenteeism based on environmental variables, which could potentially cover weather conditions in forecasting absenteeism. However, they did not include labor market trends, long-term trends in morbidities, or demographics in their model. Moreover, a model with environmental predictors would need to forecast long-term weather conditions as well, which appears difficult [22]. Jakovljević et al. [23] presented a forecasting approach for absenteeism in a selection of OECD countries, which fitted ARIMA models to a variety of economic and epidemiological summary statistics to derive the volume and costs of absenteeism in the study countries over a forecast horizon of nine years. The model, however, included demographics only indirectly as far as they were covered by the historical data for the summary statistics. A forecast model for absenteeism in the context of aging and the effect on the associated economic costs was suggested by Wilke [18], also for the German case. The model was based on age-specific morbidity risks derived from data provided by the Federal Institute for Occupational Safety and Health (BAuA) [24]. In our contribution, we further elaborate this approach by connecting it to a stochastic population forecast for Germany, developed by Vanella and Deschermeier [7]. Moreover, we take trends in labor force participation into account to acknowledge their potential impact on absenteeism. For this, we implement updated forecasts of age- and sex-specific LFPRs in our model, which have been conducted by Fuchs and colleagues based on an own stochastic forecast model [3,25]. Our model, therefore, adds to the literature by not only considering demographic, economic, and epidemiological aspects but also including past trends and quantifying stochasticity in the projections using credible intervals.

Section 2 outlines the data and the methodological approach used in our study in more detail. Section 3 shows the results for our projection of future labor force development and corresponding cases of absenteeism until the year 2050—provided disability rates remain constant and age-specific employment is as predicted by Fuchs et al. Subsequently, these purely demographic effects on the future development of absenteeism are evaluated further in terms of the associated economic costs. In Section 4, we then discuss limitations and potential improvements before we close with an outlook on questions for further research.

## 2. Materials and Methods

### 2.1. Stochastic Population Forecast

As a baseline of our projection, we computed a stochastic population forecast by age and gender for Germany over the period 2021–2050, following a stochastic cohort-component approach for age- and sex-specific population forecasting suggested by Vanella and Deschermeier [7] with an adjustment to the migration forecast. There, the authors forecast the population based on stochastic principal component-based time series methods for the major demographic components of fertility, international net migration, and mortality. Stochasticity is considered in the model by Monte Carlo simulation (10,000 draws) of all variables, resulting in 10,000 trajectories of the future population by age and gender. Our migration model, however, differs from that suggested in the mentioned study. For Germany, migration rates are more stable than flows, as they consider the baseline popu-

lation [19]. Moreover, for the case of small-area migration in Germany, Vanella et al. [26] showed that a naïve random walk forecast of pseudo migration rates gives better fits to historical migration patterns in Germany. In that vein, we used the latter approach, adjusted to national age- and sex-specific pseudo net migration rates for Germany.

Hence, in the first step of the analysis, we forecast the future birth numbers. For this, we multiplied forecasts of the age-specific fertility rates (ASFRs), as derived according to Vanella and Deschermeier [7,16], with simulations of the female population in the corresponding fertile age. In other words, in trajectory $t$, the births in year $y$ $(B_{y,t})$ are the scalar product of the vectors of the simulated ASFRs and the corresponding female population

$$B_{y,t} = \sum_{a=15}^{50} \varphi_{y,a,t} * P_{y-1,a-1,f,t} \tag{1}$$

with $\varphi_{y,a,t}$ being the ASFR of females aged $a$ years in trajectory $t$ in year $y$ and $P_{y-1,a-1,f,t}$ being the simulated female population in age $a - 1$ at year-end $y - 1$. Therefore, the ASFR is estimated on the female population at the end of the previous calendar year. As there are no clearly defined lower or upper limits of the reproductive age, 15 is defined as the lower limit, and age group 50 refers to the baseline population 50–54 years of age. Therefore, we assume the same ASFR for females aged 50–54. These assumptions, based on the available data, give plausible age schedules for fertility and, therefore, offer a good basis for forecasting the total births. The baseline birth data have been partly (1968–1991) provided on request by the German federal statistical office (Destatis) for the two former German states to the first author of the present paper for earlier studies [27–30], and partly (1992–2020) been downloaded for unified Germany from Destatis' database GENESIS-Online [31]. The denominators of the ASFRs (and all other rates used for the study) are the corresponding end-of-year population estimates, which are taken from the human mortality database (HMD) for all years until 2017. Note that the HMD estimates refer to the 1 January rather than 31 December. Therefore, for instance, the population estimates by the HMD for 2018 correspond to Destatis estimates for 2017 [32–34]. The HMD estimates derived by Klüsener et al. [35] are, especially for the years before 2011, preferable to the original Destatis data, as they are adjusted to errors arising from the population updating in intercensal periods and errors from unobserved migration, which particularly bias the old-age population estimates [7]. Since at the time of writing this paper, no data beyond 2017 were available on the HMD, we relied on original Destatis population data, provided on request to us [36], for the years 2018–2020.

We applied principal component analysis (PCA) on the logit-transformed ASFR time series, as suggested by Vanella and Deschermeier [16], using 0 as the lower and 1/7 as the upper bound. This way, we made sure that eventual simulations for the ASFRs cannot take values at exactly these bounds or beyond. The implicit assumption for the lower bound is that an ASFR cannot be zero or negative, which means we have no zero or even negative birth number in any year or trajectory for any of the defined age groups. For the upper bound, a value of 1/7 means that we assume that no more than every 7th female in the age group will bear a child in a certain year, which is approximately the historic maximum for the period since 1973 and therefore avoids trajectories with unrealistically high ASFRs. Mathematically, the logit transformed ASFR of females aged $a$ years in year $y$ is

$$l\varphi_{y,a} = ln\left(\frac{\varphi_{y,a}}{1/7 - \varphi_{y,a}}\right) \tag{2}$$

The *i*th principal component (PC) of the logit-ASFRs is then

$$PC_{y,i}^F = \sum_{a=15}^{50} \lambda_{i,a} \times l\varphi_{y,a} \qquad (3)$$

with $\lambda_{i,a}$ being the loading of the logit-ASFR of females aged *a* on PC *i* (a more detailed presentation of applied PCA in demography can be found in [37], for instance). We then fit parametric trend functions to the PC time series by ordinary least squares; the residuals between the time series and the trend fits were modeled by ARIMA models, which were specified based on graphical analysis of the time series plots, the autocorrelation functions, and the partial autocorrelation functions of the residuals. A more detailed description of the fertility model is given in Vanella and Deschermeier [16].

Second, after deriving the birth numbers, we simulated the share of males (and indirectly females), respectively, among all births (the official statistics do not differentiate further genders but distributes them to either males or females. Accordingly, our forecast sticks to the binary gender definition) by time series ARIMA methods, following Vanella and Deschermeier [7]. Gender shares among the births since 1950 were downloaded from GENESIS-Online [5]. Male births were consequently computed by multiplying the share of males in year *y* in trajectory *t* $(r_{y,t})$ with all births in the said year and trajectory:

$$B_{y,m,t} = r_{y,t} \times B_{y,t} \qquad (4)$$

The female births are, consequently,

$$B_{y,f,t} = (1 - r_{y,t}) \times B_{y,t} = B_{y,t} - B_{y,m,t} \qquad (5)$$

Third, we simulated migration in- and outflows by age and gender, generalizing the suggestion by Vanella et al. [26]. Migration flows are not available by cohort but only completed years of age of the migrants. We, therefore, assumed migration and births to occur uniformly over the year, which in praxis is not the case. For births, for instance, there is strong annual seasonality [38]. These assumptions allowed us to approximate cohort-based age- and sex-specific migration flows based on the migration flows by completed age through averaging. In other words, based on the assumption that 1/12 of all annual migrations happen each month, we can conclude that half of the annual migrations have happened by the end of each year. Therefore, the average age of a migrant is exactly between two adjacent birthdays. For instance, among female migrants of age 30 in the year 2020, one-half are assumed to have been born in the calendar year 1990, while the other half consequently has been born in 1989. Based on these flow data, we computed age- and sex-specific emigration rates and pseudo-immigration rates, following Fuchs et al. [19]. Migration flows by age and gender for the years 1991–1999 had been provided to the first author of the present paper by Destatis for an earlier study [39], the corresponding data for the period 2000–2020 are available via GENESIS-Online [40]. Based on the migration flow data in connection to the earlier mentioned population estimates, we derived (pseudo-) migration rates, which we forecast according to Vanella et al. [26]. The (immigration) emigration of individuals of age *a* and gender *g* in year *y* and trajectory *t*, therefore, is given by

$$I_{y,a,g,t} = \begin{cases} i_{y,0,g,t} \times B_{y,g,t} \; for \; a = 0, \\ i_{y,a,g,t} \times P_{y-1,a-1,g,t} \; for \; a > 0 \end{cases} \tag{6}$$

and

$$E_{y,a,g,t} = \begin{cases} e_{y,0,g,t} \times B_{y,g,t} \; for \; a = 0, \\ e_{y,a,g,t} \times P_{y-1,a-1,g,t} \; for \; a > 0 \end{cases} \tag{7}$$

for immigration and emigration, respectively. Here, $i_{y,a,g,t}$ ($e_{y,a,g,t}$) is the pseudo-immigration (emigration) rate of individuals aged $a$ of gender $g$ in year $y$, and trajectory $t$. Note that migration estimates here are computed as the product of the pseudo-immigration (emigration) rate and the end-of-year population of the corresponding cohort in the previous year and the target (origin) country. To include migration among newborns, we referred to the births occurring in year $y$ as the denominator. Following Vanella et al. [26], we computed pseudo-net migration rates by age and gender as the difference of the pseudo-immigration rate and emigration rate of the same stratum, i.e., $n_{y,a,g,t} := i_{y,a,g,t} - e_{y,a,g,t}$. Vanella et al. [26] showed that for Germany, a naïve multivariate random walk approach based on migration rates performs best to predict future migration flows. We adjusted their approach by assuming the first differences of the pseudo-net migration rates $\Delta n_{y,a,g,t} := n_{y,a,g,t} - n_{y-1,a,g,t}$ to follow a multivariate Gaussian distribution:

$$\begin{bmatrix} \Delta n_{y,0,m,t} \\ \Delta n_{y,1,m,t} \\ \vdots \\ \Delta n_{y,0,f,t} \\ \vdots \end{bmatrix} \sim \mathcal{N}\left(\vec{0}, \Sigma\right) \; \forall \; y, t \tag{8}$$

with $\vec{0}$ being a 2a-dimensional (in our case 192-dimensional) null vector and $\Sigma$ being the empirical covariance matrix of the first differences of all age- and sex-specific pseudo-net migration rates for 1995–2020. Monte Carlo sampling from (8 leads to 10,000 trajectories of the age- and sex-specific pseudo-net migration rates. Multiplying those with the population bases according to (6) and (7), we can simulate the distributions of net migration for each stratum and year as

$$N_{y,a,g,t} = \begin{cases} n_{y,0,g,t} \times B_{y,g,t} \; for \; a = 0, \\ n_{y,a,g,t} * P_{y-1,a-1,g,t} \; for \; a > 0 \end{cases} \tag{9}$$

Fourth, we computed the deaths according to Vanella and colleague [7,17]. For this, age- and sex-specific survival rates (ASSSRs) were computed retrospectively as quotients of survivors of some cohort at the end of the year and the sum of the survivors and deaths among the same cohort at the same point in time. This way, we included mortality among the children born in the current year and migration into our mortality estimates, including the timing of births and migration as represented by the historical data. We forecast mortality by forecasting logit-ASSSRs (in this case using the standard logit with 0 as the lower and 1 as the upper bound) based on PCA and fitting trend functions and ARIMA models to PC times series, similarly as explained for the case of ASFRs. A more detailed presentation of the ASSSR forecast model is given by Vanella [17]. The simulation of survival rates combined with the population estimates for the previous year and the migration flows and births simulated according to (5)–(9) gives the death simulations:

$$D_{y,a,g,t} = \begin{cases} \left(1 - s_{y,0,g,t}\right) \times B_{y,g,t} \; for \; a = 0, \\ \left(1 - s_{y,a,g,t}\right) \times \left(P_{y-1,a-1,g,t} + N_{y,a,g,t}\right) \; for \; a > 0 \end{cases} \tag{10}$$

with $s_{y,a,g,t}$ being the survival rate of individuals of age $a$ and gender $g$ in year $y$ and trajectory $t$. The reciprocal $\left(1 - s_{y,a,g,t}\right)$ is the corresponding mortality rate. The data underlying the computation of the survival rates are, next to the earlier mentioned population

estimates, data on annual deaths by cohort and gender, provided on several occasions on request by Destatis to the first author [41,42].

Fifth, combining the results from (5)–(10), we obtained the cohort-component simulations of the end-of-year population as

$$P_{y,a,g,t} = \begin{cases} B_{y,g,t} + N_{y,0,g,t} - D_{y,0,g,t} \ for \ a = 0, \\ P_{y-1,a-1,g,t} + N_{y,a,g,t} - D_{y,a,g,t} \ for \ a > 0 \end{cases} \tag{11}$$

*2.2. Projection of Labor Force Participation in the Context of Increasing Retirement Ages*

Sixth, we combined our population forecast obtained from (11) with an updated stochastic forecast of LFPRs suggested by Fuchs et al. [3], which includes trends in age-, gender- and nationality-specific labor market participation. For instance, the LFPR of individuals of age *a* and gender *g* in year *y* $(l_{y,a,g})$ represents the share of individuals in the said stratum available to the labor market, relative to the population of the same age and gender. This does not directly include the demand for labor or how many of these individuals actually will have a job but simply the labor supply based on the willingness to take on employment, taking historical trends in LFPRs by age, gender, and nationality (Germans or non-Germans) into account. These forecasts are especially crucial for our research question, as trends of increasing LFPRs are observable in the long run among females [3], and we will see increasing LFPRs among the elderly as a consequence of demography-related pension reforms, which cause legal retirement ages to increase until 2031 [6]. The updated LFPRs are provided by Fuchs et al. [25]. The authors, therefore, include trends in labor force participation by migration (for instance, longer periods of labor market integration) in their forecast. Therefore, we implicitly assume shares of migrants in the population according to Fuchs et al. [25], as our model does not predict the foreign separately from the German population. The projection of labor supply by age group and gender can then be derived by

$$L_{y,a,g,t} = l_{y,a,g} \times P_{y,a,g,t} \tag{12}$$

with $l_{y,a,g}$ being the median forecast of the LFPR of individuals of age *a* and gender *g* for year *y* according to Fuchs et al. [25]. Note that we included the LFPRs only deterministically in the model, as they are subject to many aspects, such as the socioeconomic composition of the population or the overall economic development. Since our focus was to investigate the pure demographic effect on future absenteeism, a deterministic inclusion of labor market effects fitted our purpose. Including at least the expected trends in labor force participation appeared necessary, however, to give a realistic estimate of the demographic effect under increasing retirement ages. We only adjusted the LFPRs insofar, as Fuchs et al. [25] simulated five-year age groups. Our model is based on single years of age, and we wanted to avoid hard cuts in the LFPR curves, which are a consequence of larger age groups. Therefore, we assumed age- and sex-specific LFPRs as provided in Fuchs et al. [25] for age groups for the median age of each age group (e.g., for age group 25–29, the LFPR from Fuchs et al. is assumed for age 27 in our model), and interpolated the LFPRs for the ages between the knots by natural cubic splines [43]. Yet, our aim is not to give a forecast of the future labor supply but to include the expected effects of labor supply change in our model of future absenteeism. Since labor supply does not equal the population in labor, the labor force projections are just used as auxiliary data that include the effect of delayed retirement in our projection. Assuming that the share of the labor force that is employed remains constant, the relative change between periods *y* and $y + \tau$ in employed persons for each age-gender stratum under changing LFPRs and the demographic trends in trajectory *t* derived from (12) can then be computed by

$$\ell_{y,\tau,a,g,t} := \frac{L_{y+\tau,a,g,t}}{L_{y,a,g,t}} \tag{13}$$

## 2.3. Projection of Relative Increase in Absenteeism Given Demographic and Economic Trends

Seventh, we combined our projections of the labor force effect according to (13) with information on absenteeism by age group to deliver stochastic estimates of the demographic effect on absenteeism. We do not know the exact number of days of absenteeism. The most informative data available publicly are annual reports on work-related safety and health issues, provided by the Federal Ministry of Labor and Social Affairs (BMAS) and the BAuA. We specifically used the latest report for the year 2019 [15]. In Germany, every individual is health insured by either social health insurance (GKV) or private health insurance (PKV) companies. In our reports, the data are restricted to GKV insured; age-specific data for PKV insured are not publicly available. We used estimates for both cases of absenteeism, standardized to 100 years of insurance membership, and averages of days per case are reported for 5-year age groups. The numbers are given in Table 1.

**Table 1.** Standardized age-specific cases of work absence per one year of full insurance and average length of absence as days for GKV members in 2019 (Sources: [15]; authors' computation and illustration).

| Age Group | Annual Cases per Capita | Average Days per Case | Average Annual Days per Capita |
|---|---|---|---|
| 15–19 | 2.57 | 5 | 12.85 |
| 20–24 | 2.1 | 6 | 12.6 |
| 25–29 | 1.65 | 8 | 13.2 |
| 30–34 | 1.6 | 9 | 14.4 |
| 35–39 | 1.6 | 10 | 16 |
| 40–44 | 1.53 | 11 | 16.83 |
| 45–49 | 1.48 | 13 | 19.24 |
| 50–54 | 1.53 | 15 | 22.95 |
| 55–59 | 1.65 | 17 | 28.05 |
| 60–64 | 1.74 | 21 | 36.54 |
| 65+ | 0.71 | 23 | 16.33 |

The number of cases does not increase monotonically by age, as knowledge about the connection between age and morbidities would suggest (see, for instance, [13] on disability risks in the context of long-term care insurance). Instead, we have more of a bathtub shape in the age groups below 65, followed by a sharp decrease for the elderly. The average sick days per case, however, show an increasing duration of absenteeism with increasing age. Multiplying these two statistics gives an average of days of absenteeism per year, standardized to one year of full insurance as an employed person in the GKV, which is given in the last column of Table 1. There, we see clear trends of longer absenteeism for age over 24. For the teenage group, we see a slightly higher average absenteeism in comparison to the 20- to 24-year-olds. This might be a rounding error resulting from the rounding in the available data (note that the average of days per case is rounded to integers). An alternative explanation might lie in more absenteeism as a result of more injuries due to riskier behavior, e.g., observable in higher traffic accident rates [44]. The dip in the elderly is unintuitive but could be explained by a positive selection of healthier workers employed in less exhaustive fields and having a higher work motivation, therefore, in many cases working beyond their individual retirement age, following Wilke [18].

It has to be noted that the data are not just a sample of all insured workers but somewhat biased. Age structure [45] and morbidities of the insured diverge, with the PKV attracting, on average, wealthier and healthier customers [46], since insurance companies offer more individualized contracts [47], and in some regions, a more extensive healthcare supply is available for PKV in comparison to GKV insured [48]. Therefore, our data will probably overestimate the morbidities of the overall population. To circumvent this limitation in the data, we did not directly address the days of absenteeism but simply assume the relation of sick days between the age groups, i.e., the risk of sickness, to remain

constant over time. Moreover, we assumed constant shares of members in the GKV and PKV for the projection. Under these assumptions, we can project the relative change in absenteeism over time, restricted to future developments in the population size and structure and labor force participation, including increases in incidences in the oldest age group caused by increases in legal retirement ages which will occur until 2031 [6]. Yet again, we took the age group-specific numbers from Table 1 and interpolated them by natural cubic splines, as performed for the LFPRs. Our sickness data does not discriminate by gender; therefore, we assumed the morbidities to depend exclusively on age, not gender. Indeed, differences in morbidities between the genders exist, yet appear more subtle in the labor age groups, becoming more crucial among the elderly [13,49]. Therefore, the error resulting from this assumption appears negligible. In particular, we took (13), which includes demographic and economic trends and multiplied with average annual per capita absenteeism $(d_{y,a})$, which then serve as some kind of weighting factor of age-related morbidity:

$$S_{y,a,g,t} := d_{y,a} \times L_{y,a,g,t} \tag{14}$$

where $S_{y,a,g,t}$ is the annual sick days of individuals of age $a$ and gender $g$ in trajectory $t$ and in year $y$, which includes both demographic effects (stochastic) and labor market effects (deterministic), applying a status quo assumption to epidemiological trends. As $d_{y,a}$ is biased [18], we opted to derive the relative change in absenteeism instead of absolute numbers. Assuming that the relative bias of age-specific absenteeism is the same for all age groups, the biases will cancel out in a relative measure. The sum of absenteeism in year $y$ over all demographic groups is $S_{y,t} = \sum_a \sum_g S_{y,a,g,t}$. The absenteeism in year $y$, relative to the year 2020, in each trajectory is then

$$\kappa_{2020,y,t} := \frac{S_{y,t}}{S_{2020}} \tag{15}$$

### 2.4. Projection of Relative Increase in Economic Costs by Absenteeism Trends

In the eighth and final step, we projected how, based on our previously derived results, absenteeism will affect health economics. Given the derived demographic development and combined with the assumed economic and epidemiological trends, we projected costs caused by productivity loss, ceteris paribus. For this, we borrowed Wilke's [18] estimates of age-specific productivity, measured in EUR of 2018 and loss thereof for each day of absenteeism. The most important numbers are given in Table 2.

**Table 2.** Annual gross income and productivity loss per day of absenteeism 2018 (Source: [18]; own illustration).

| Age Group | Average Gross Income [as €] | Loss of Productivity by Day [as €] |
| --- | --- | --- |
| 20–24 | 21,246 | 58.21 |
| 25–29 | 31,790 | 87.10 |
| 30–34 | 39,826 | 109.11 |
| 35–39 | 43,083 | 118.04 |
| 40–44 | 45,610 | 124.96 |
| 45–49 | 46,075 | 126.23 |
| 50–54 | 45,972 | 125.95 |
| 55–59 | 43,689 | 119.70 |
| 60–64 | 40,853 | 111.93 |
| 65–69 | 16,233 | 44.47 |

Again, we take the last column and interpolate the values as described earlier. We can then estimate the relative change in economic costs of absenteeism by multiplying (14) with the average daily loss of productivity for the corresponding demographic stratum, say $c_{y,a}$:

$$C_{y,a,g,t} := c_{y,a} \times S_{y,a,g,t} \tag{16}$$

Similar to (15), we compute the change in economic costs by absenteeism, relative to 2020 by

$$\pi_{2020,y,t} := \frac{C_{y,t}}{C_{2020}} \tag{17}$$

with $C_{y,t}$ being the total costs of absenteeism calculated over all age groups in year $y$ and trajectory $t$.

We will present the results of our simulations in the next section.

## 3. Results

Figures 1 and 2 illustrate the results of our population forecast. We show the forecasts until 2050, aggregated to 10-year age groups. As our focus was on the labor age population, we limited our analysis to age groups 15–74 (the model simulates ages by year 0–99, and 100+ as a cumulative age group). The black continuous lines show the past observations since 1990, the median forecast is visualized as blue dashed lines, and the 75% prediction intervals (PIs) are added as violet dotted lines. Those were derived from Monte Carlo simulation via cohort-component forecasting as described in Section 2.1. Interested readers can find forecast results for the demographic components in Appendix A.

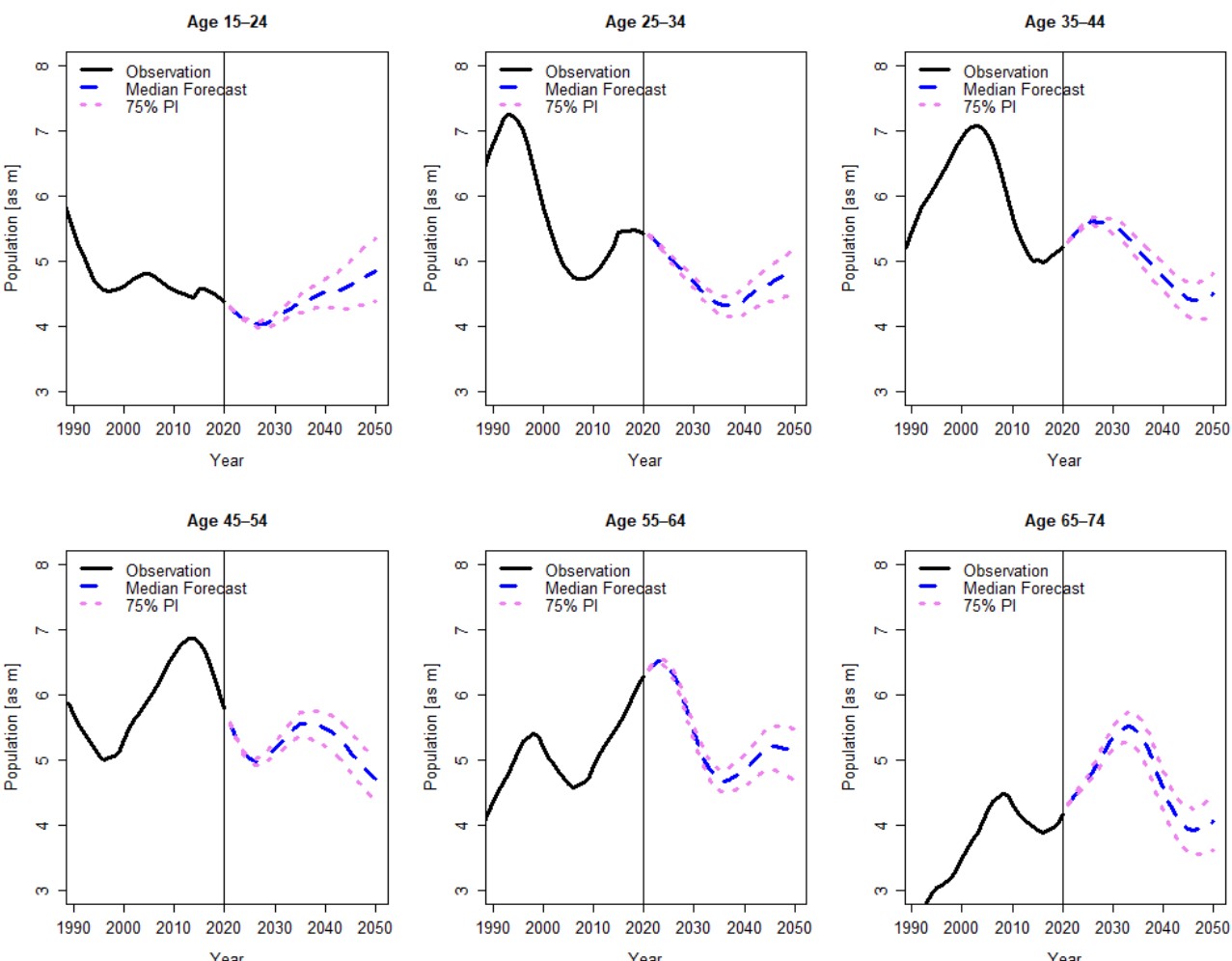

**Figure 1.** Forecasts of male population in labor age by 10-year age groups with 75% prediction intervals (Sources: [33]; authors' computation and illustration).

We expect a dip in the younger population by the end of the current decade, which is a consequence of a long period of very low fertility rates in Germany (the total fertility rate in Germany has consistently been below 1.5 children between 1981 and 2014 [16]).

As a consequence of increasing fertility rates and positive net migration in the young age groups, this trend will invert in the long term, however. Volatility in the youngest age groups increases significantly at the end of the 2030s, which is because a large share of that age group is not yet born. An apparent trend is a large wave we see in the age group 55–64 in the 2020s and, correspondingly, for the age group 65–74 in the following decade, which is the strong baby boom cohort, whose labor force participation is of special interest for our research question. The wave is more distinct for the females than for the males, which is due to the lower mortality of females, which becomes more emphatic in those age groups [17].

The latter point stresses the importance of labor force participation trends by females. LFPRs of females traditionally are, on average, smaller than for males, as the females have been taking over more responsibilities in the household, and childcare opportunities have not been sufficient to allow for both parents to participate in the labor market. Since the late 1970s, more females have had the opportunity to follow a career of their own [16]. As a consequence, LFPRs have been increasing, which is a trend expected to persist in the future [3]. Figure 3 shows the LFPRs by age group and gender, nowcast by Fuchs et al. [25], for 2020 against their median trajectory for 2050. Our cubic splines interpolations are included in the figure as lines. The authors forecast increases in the LFPRs, especially among females, under increasing trends in tertiarization and corresponding longer periods of education [50] that are connected to later entries in the labor market.

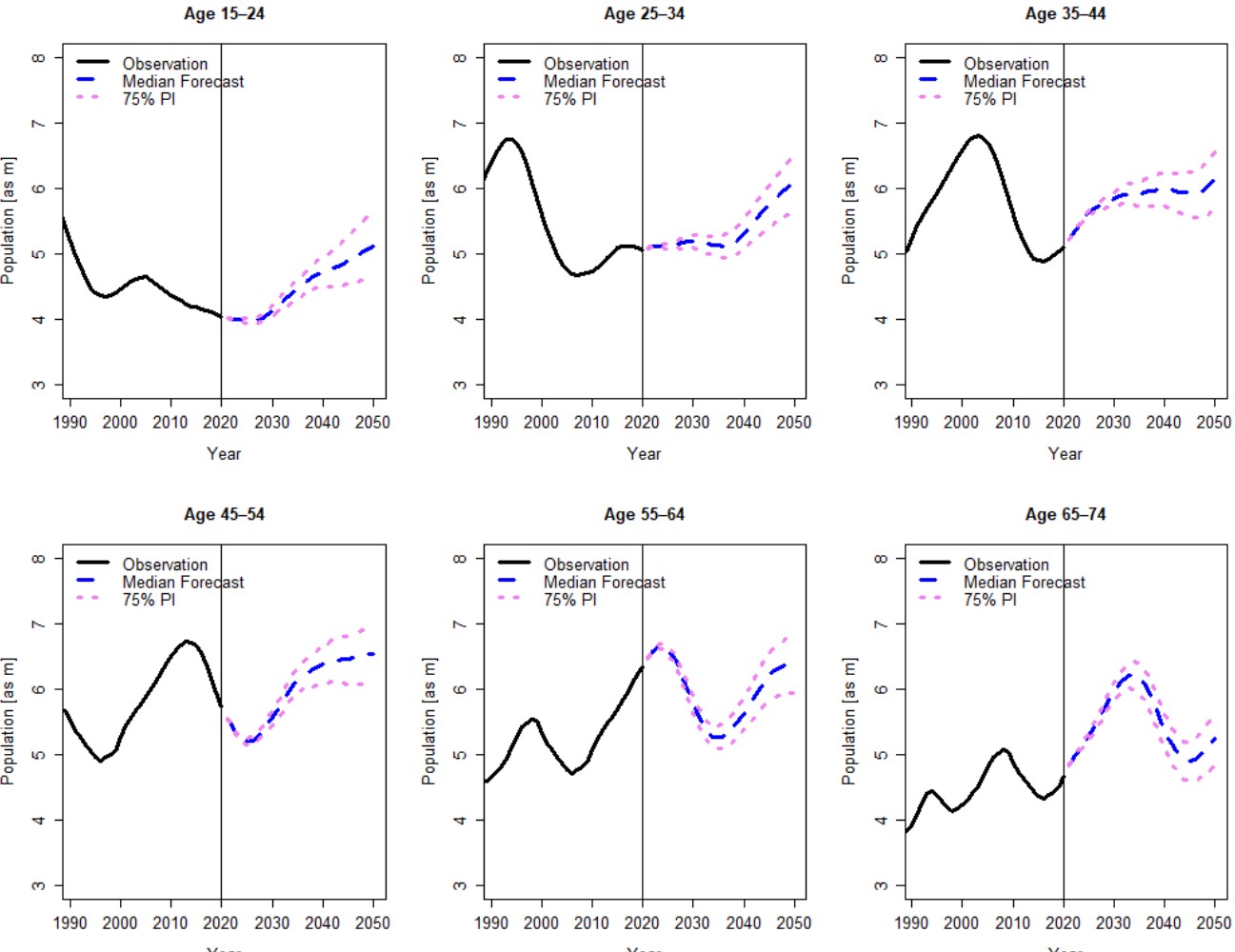

**Figure 2.** Forecasts of female population in labor age by 10-year age groups with 75% prediction intervals (Sources: [15]; authors' computation and illustration).

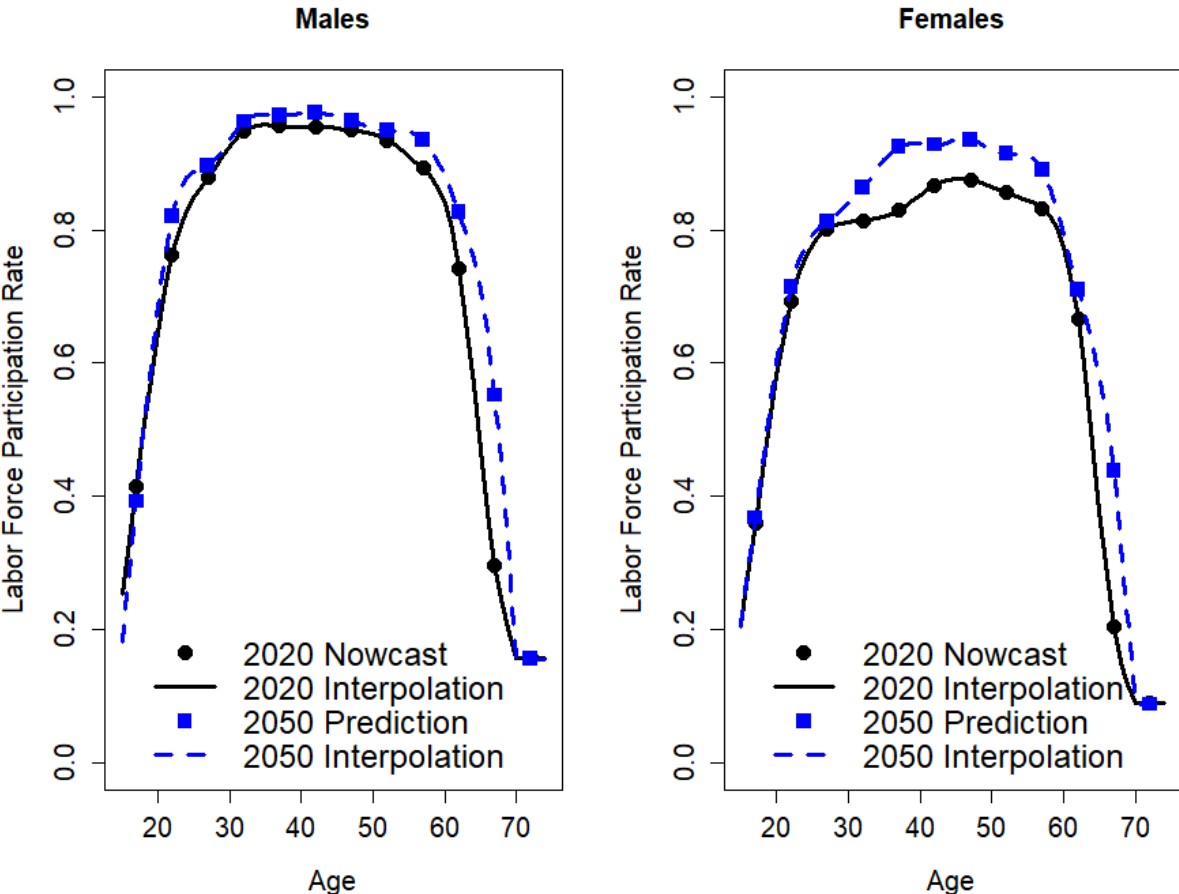

**Figure 3.** Age- and sex-specific labor force participation rates for 2020 and 2050 (Sources: [25]; authors'
computation and illustration).

Figure 4 shows our projection of the percentage change in overall absenteeism in
Germany in relation to 2020, derived from our model, with 75% PIs. It has to be stressed
that stochasticity in the model draws from demography only. Labor force participation is
assumed as illustrated in Figure 3, and prevalence rates are assumed constant as derived
by Wilke [18] and our interpolation of her estimates. Therefore, we exclusively estimate
risk arising from the demographic developments, assuming fixed trends in labor force
participation and epidemiology. Interestingly, we can expect waves in absenteeism with
increases over the next years, as the baby boomers will be in the late phase of their employ-
ment period, which is associated with relatively high morbidity and connected phases of
absenteeism (see Table 1). This development is enhanced by the mentioned increases in
legal retirement ages until 2031. The delaying retirement effect of those increases will then
stabilize, which will lead to further increases in old-age pensions [6], whereas decreases in
absenteeism can be expected. During the 2040s, increases in absenteeism can be expected as
a consequence of the high net migration Germany has witnessed since the 2010s, especially
among young migrants (see, for instance, Figures A3 and A4). Those having immigrated
at a young age in the strong migration waves will in the 2040s reach their late labor age,
which will be associated with high prevalence of absenteeism.

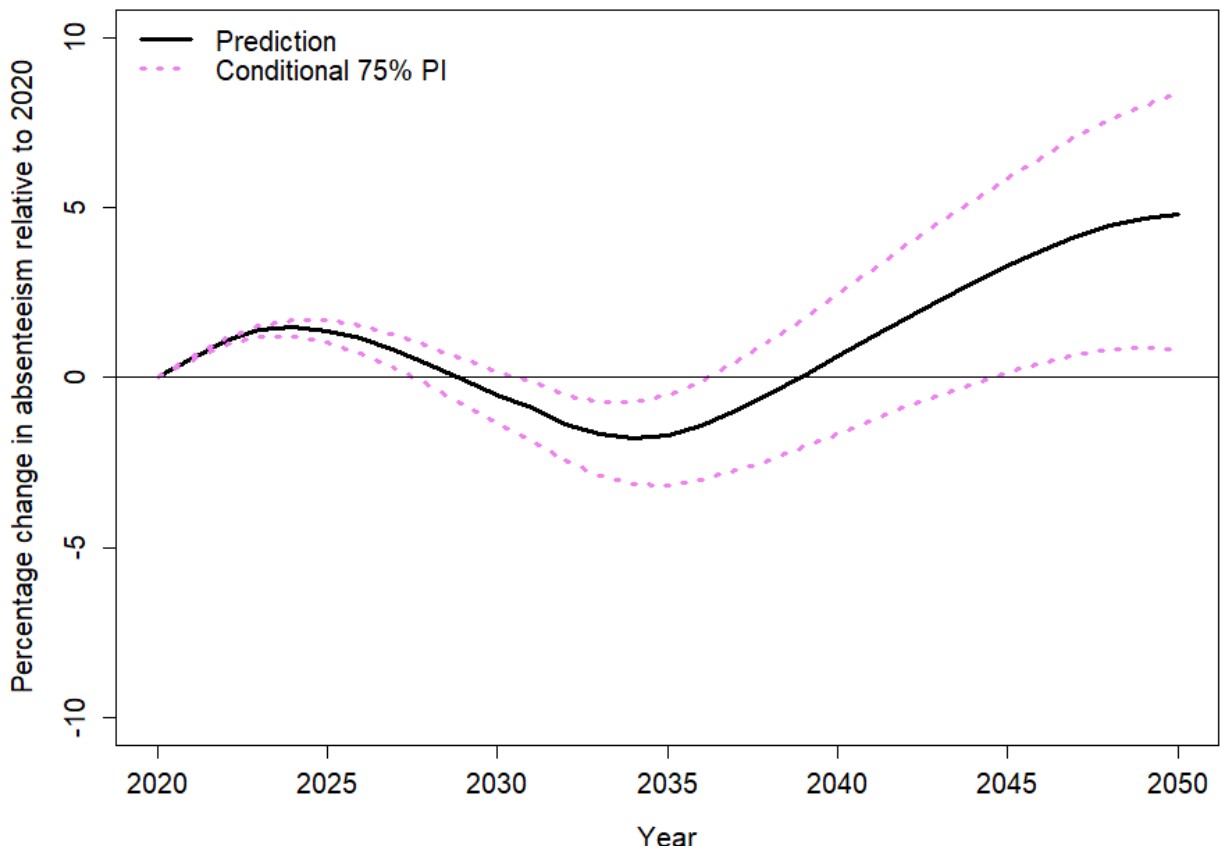

**Figure 4.** Projection of relative percentage change in overall annual absenteeism to reference year 2020 (Sources: [15,25]; own computation and illustration).

Finally, we consider the relative change in economic costs due to absenteeism associated with the demographic development, illustrated in Figure 5. Whereas the overall trending of the curve looks similar to Figure 4, we see less distinct waves, as the overall days of absenteeism factor not only into this but also the loss of productivity, measured according to Table 2. This, therefore, appears to give a more realistic picture of the economic consequences of the aging process, since we account for the worth of an employee to their employer, not just if they are at work or not.

For instance, we see that the wave between the 2020s and 2040 is, relative to the rest of the curve, much more stressed in the costs than it is in the absenteeism curve. This is because the high decrease in absenteeism is very much caused by the retirement of the baby boomers, who will have been causing relatively high productivity losses until the mid-2020s, shortly before their retirement, making them vulnerable to high absence rates. Since absenteeism among the labor population aged 35 and above is associated with relatively high productivity losses due to increased income (see Table 2), absenteeism will be relatively costly during that period. High retirement rates since the late 2020s will then be associated with sharply decreasing "expensive absenteeism" due to illnesses of expensive workers. Eventual increases in absenteeism costs are, in comparison, less stressed compared to days of absenteeism, as increases in absenteeism are to a higher degree associated with increasing labor force participation by females and the elderly population (see Figure 3), i.e., more absenteeism among very young and old workers, who are both less expensive (see Table 2) and therefore cause lower relative productivity losses in case of absenteeism.

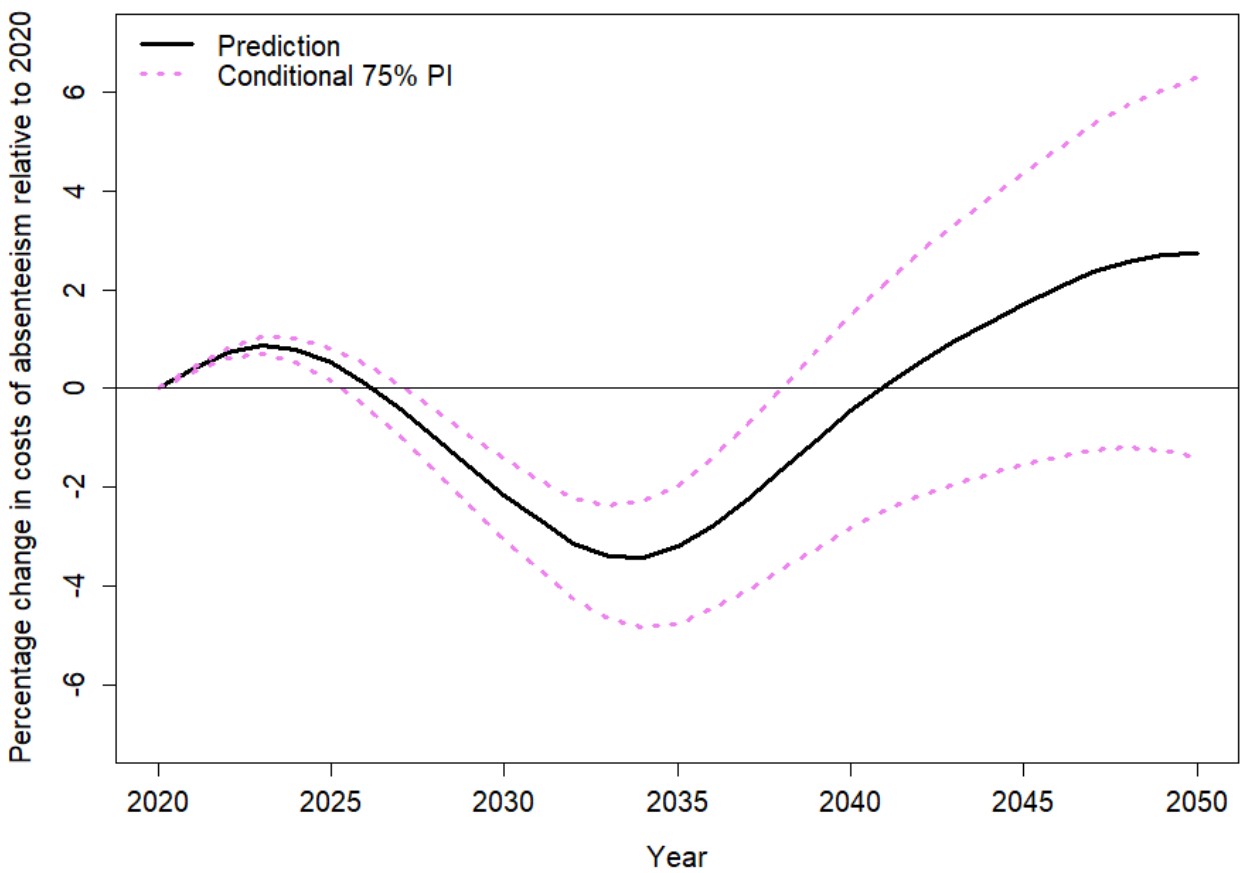

**Figure 5.** Projection of relative percentage change in overall annual costs of absenteeism to reference year 2020 (Sources: [15,18,25]; own computation and illustration).

## 4. Discussion

In this study, we developed a new approach to forecasting absenteeism and its associated costs in the context of population aging. According to our literature review, there are very few previous studies that elaborate forecast models for absenteeism, although the link between absenteeism and aging has been investigated well in the international literature for decades. There is, however, thus far only a small number of studies that provide a framework for absenteeism forecasting, and almost none give long-term forecasts for absenteeism. However, long-term trends are particularly important since they provide information for strategic planning. Wilke [18] presented a first approach to project absenteeism and its costs for the German economy, taking demographic trends and age-specific morbidity into account. However, her approach was purely deterministic and did not yet factor in trends in labor force participation. These are of particular importance, though, given the context of demography-related pension reforms and increasing female labor force participation that will affect labor supply significantly. The expected increases in absenteeism are very likely to especially affect the more vulnerable age group above age 60. Our model is not only the first that takes demographic trends stochastically into account, including time trends in the demographic variables and cross-correlations in the trends between different demographic strata, in a long-term projection of absenteeism but also the first that considers labor force participation in absenteeism forecasting, which is a yet under-investigated strand of the academic literature.

Still, our model has some limitations. First, the model relies on a forecast of the labor supply potentially available to the labor market based on a stochastic population forecast in combination with a forecast of labor force participation rates. This, however, does not necessarily correspond to the number of persons being employed as we do not model the demand side of the labor market. To do this, another sophisticated forecast of the future economic situation would be required, which is not only beyond the scope of this paper as such but also difficult to project in the long run. Moreover, additional data on the education and training of the working-age population would be necessary and would need to be matched with projected labor demand.

In addition, our projections of days of absenteeism and associated economic costs are not to be mistaken as true forecasts (for a distinction between forecasts and projections, see, for instance, [10]), as we provide forecasts on the population but include labor force participation rates deterministically only.

Moreover, we do not conduct a forecast of the relevant epidemiological parameters, i.e., we do not make any assumptions about how age-specific absenteeism might develop in the future. In an aging population, it remains unclear whether overall morbidity will decrease or increase. In the literature, two opposing theories can be found: the theory of morbidity compression and the medicalization theory. While the first theory assumes morbidities to be delayed to later points in time, quasi-parallel to increases in life expectancy [51], the latter assumes increases in life expectancy to be completely spent in poor health [52]. While there is some evidence that the truth lies somewhat in between these two extreme scenarios [13], little is yet known about age-specific morbidity patterns. We tried forecasting the epidemiological trends ourselves in gathering time series data for age-specific absence rates from all previous BMAS/BAuA reports. However, we found significant structural breaks in 2016, not allowing a sound basis for future projections. Even in the four years after the structural break (i.e., 2016–2019), the trends are not clear (e.g., the average absence days of individuals aged 60–64 years in data jumps from 30.03 to 34.86 days from 2015 to 2016, after which they increase to 37.17 days by 2018, before decreasing to 36.54 days in 2019; similar trends are observed for other age groups). Therefore, we decided to follow a simple but well-defined projection approach where epidemiological risks are assumed to remain constant. This approach allows for better isolation of the pure demographic effects without imposing potentially misleading epidemiological trends that would be based on too short and unrepresentative time series. Longer and more consistent time series will allow deriving epidemiological trends from the data as well, which will be a potential improvement on our model. Changes in absence risks in the model are thus exclusively rooted in changes in age-specific LFPRs and therefore only depict the pure demographic effects.

Regarding days of absenteeism and associated economic costs, our data were restricted to GKV data, which might be strongly biased, as has been discussed earlier. To account for this, we do not deliver absolute results for days of absenteeism and associated economic costs but only relative changes over time so that biases level off.

Last, our model only relies on prepandemic data. Therefore, our forecast does not include any influence the COVID-19 pandemic might have on demographic or epidemiological trends. For our model, however, it is only the working-age population that is relevant. Vanella et al. [53,54] showed in international studies on excess mortality and case fatality risks that mortality trends in these age groups so far have not been significantly tampered with by COVID-19. Therefore, the underlying mortality forecast is robust. For fertility in Germany, Vanella et al. [38] did not find statistical evidence for any effect of the pandemic. Therefore, our birth forecast holds as well. International migration flows slumped during the pandemic. So far, however, the effect of the pandemic on international migration is unpredictable as reliable data is still missing. The pandemic in 2020 halted the rise in employment rates that has been continuing since 2006. Even though LFPRs collapsed in the first months of the pandemic, a rising trend is anticipated again once the pandemic situation has ended. A similar trend is expected for 2021 [55]. Germany introduced the

first major contact reduction measures in March 2020 [56]. Overall, the short-term effects of the pandemic can be explained quite well. For example, well over half of the significant increase in transitions from employment to unemployment in April 2020 was probably due to the shutdown. The effects were even more pronounced in the absence of new hiring [57]. It is not yet clear, however, whether and to what extent long-term negative effects of the pandemic are to be expected. With or without the pandemic, in the medium and long run, a decline in the working-age population will be inevitable given the German demographic development [3].

### 5. Conclusions

We have suggested a forecast model of absenteeism, using Germany as a case study and projecting relative change in absenteeism both in days and costs until 2050. We showed that absenteeism is a complex and, for aging economies, severe topic. Yet, it is rather under-investigated in the scientific literature. Absenteeism depends not only on demographic development but also on socioeconomic trends, such as labor force participation, as well as epidemiological trends, such as trends in morbidity prevalence and working environments. We focused on the demographic part in the most detail, employing an adjusted version of an established stochastic population forecast model for Germany, suggested by Vanella and Deschermeier [7], including labor force participation rates and morbidity rates deterministically.

LFPRs are assumed according to a recent forecast by Fuchs et al. [25], whereas age-specific morbidity rates are held constant as under the status quo scenario. We believe that our model thus offers a new approach toward research in economic and demographic forecasting with great potential for further development and improvement.

A higher differentiation of absenteeism, e.g., by gender or nationality, would improve our projections further. However, this type of data is not publicly available. It would be important to generate consistent time series of absence rates for this, however. The available data show significant structural breaks, presumably caused by changed definitions in the data over time. We did, therefore, assume age-specific morbidities to remain on a stable level, which has a high potential for error. However, we preferred this assumption against extrapolating trends from unrepresentative time series data or a theory-based simulation, which has been suggested earlier. Instead, we confirm this rather simple assumption, which the reader should, however, keep in mind when considering our results. Longer time series of age-specific absence rates without structural breaks due to definition or data problems would further benefit our approach, as we could generate time trends in morbidities from these—so far, these data are not publicly available, though.

Keeping these limitations in mind, our model provides a solid first step toward understanding absenteeism and its economic costs given the ongoing aging process. Furthermore, we show that increases in the statutory retirement age do not lead to one-to-one increases in LFPRs but are also, as shown by Vanella et al. [6], associated with increases in disability pensions and higher absenteeism. Common projections on demography and pensions typically do not take these aspects into account (see, for instance, Vanella et al. [6], for an overview). Our approach, therefore, could be applied to projections of future labor market and pension development as a baseline for economic planning and political decision-making. Whereas our study focuses on the case of Germany, the method can be applied to other countries or larger geographical units such as the EU as underlying models, namely forecasts for the demographic size and structure of the working-age population, labor force participation rates, and absence rates can be generated or are available for further use. For instance, Vanella et al. [10] provided an overview of different techniques for population forecasting with indications for further reading, also naming some selected specific sources for global as well as national population projections that could be used as a basis for working-age population projections. LFPRs can be projected using a method similar to that suggested by Fuchs et al. [3]. Global relevant labor market data, such as demographically stratified LFPRs, are provided by the International Labour Organization [58]. Morbidity rates can

either be derived by national surveys or be assumed similar to the values for Germany reported in Table 1. After all, the method can be adjusted to the data available for any specific country or region. If projections and representative time series data are missing to derive robust trends in LFPRs, model assumptions can be plugged into the model under the limitations arising from these assumptions. In our case, e.g., we assume age-specific absence rates to remain constant due to a lack of representative time series data. Countries with valid time series data on absence rates could instead derive long-term trends as well. To conclude, our model framework thus also can easily be applied to other regions of the world.

**Author Contributions:** Conceptualization, C.B.W. and P.V.; methodology, P.V., C.B.W. and D.S.; software, P.V. and D.S.; validation, all authors; formal analysis, P.V. and D.S.; investigation, C.B.W. and P.V.; resources, P.V. and D.S.; data curation, P.V. and D.S.; writing—original draft preparation, C.B.W. and P.V.; writing—review and editing, P.V.; visualization, P.V.; supervision, C.B.W.; project administration, P.V. All authors have read and agreed to the published version of the manuscript.

**Funding:** This research received no external funding.

**Institutional Review Board Statement:** Not applicable.

**Informed Consent Statement:** Not applicable.

**Data Availability Statement:** Data generated by our study are provided by the corresponding author on reasonable request.

**Acknowledgments:** We thank Johann Fuchs for his valuable advice and consultation in the labor force part of the model. Moreover, we appreciate the timely and helpful remarks by the three anonymous reviewers, which have led to improvements in the manuscript.

**Conflicts of Interest:** The authors declare no conflict of interest.

## Appendix A. Further Results

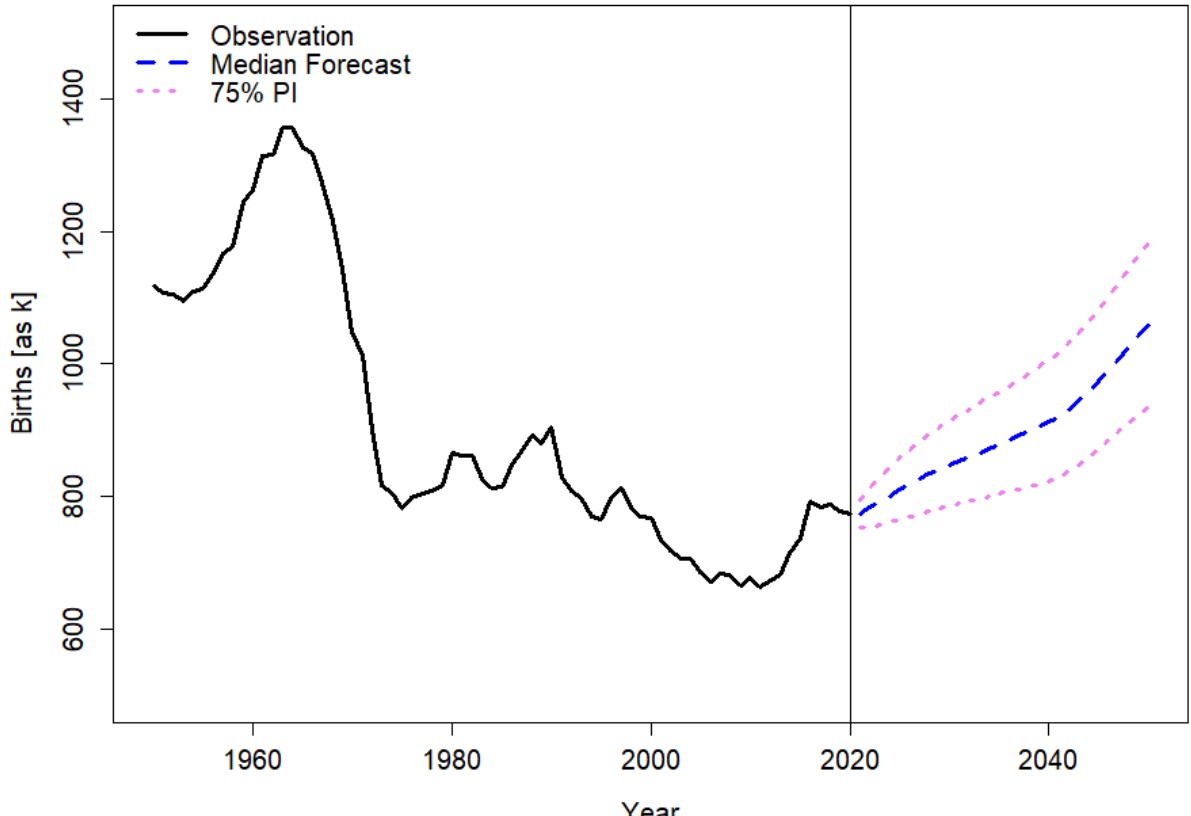

**Figure A1.** Births forecast with 75% prediction interval (Sources: [19]; authors' computation and illustration).

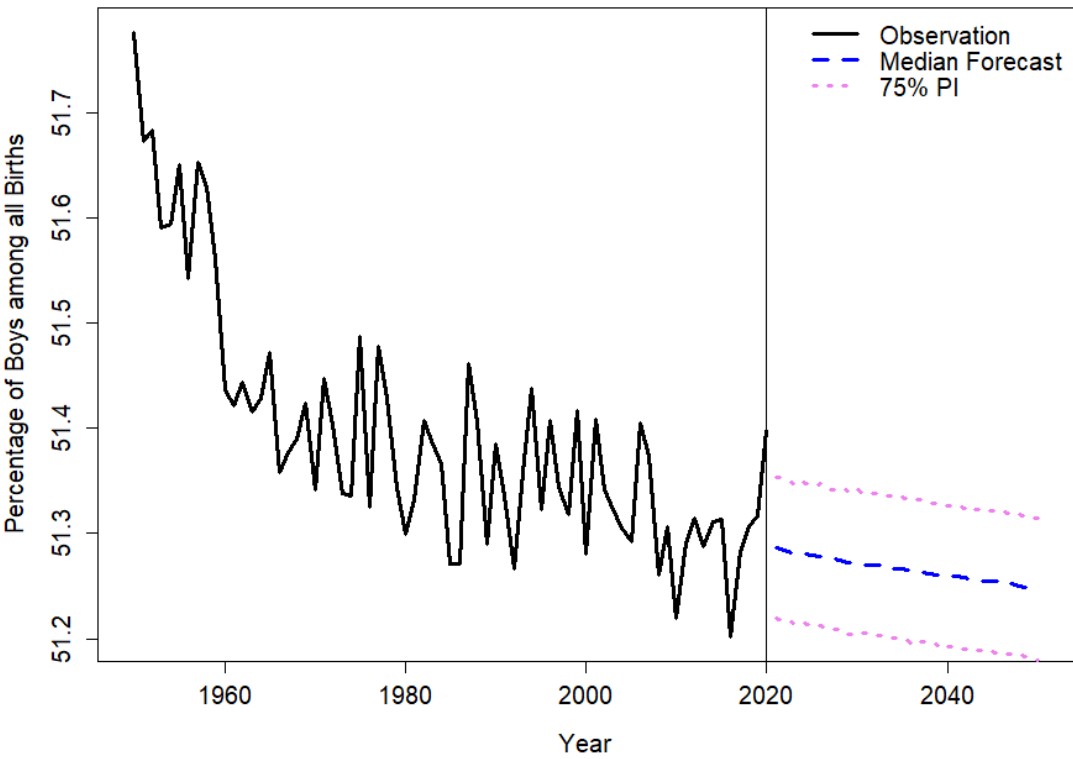

**Figure A2.** Forecast of males among all births with 75% prediction interval (Sources: [19]; authors'
computation and illustration).

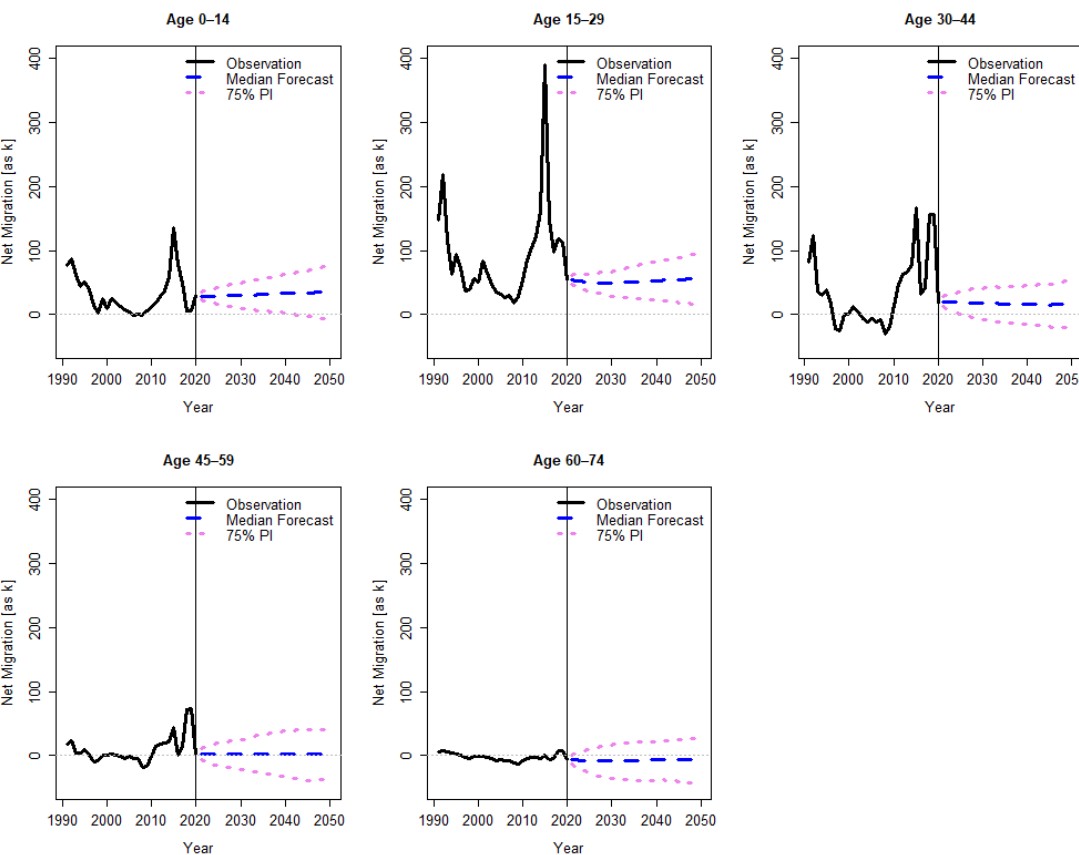

**Figure A3.** Forecasts of net migration of males by 15-year age groups with 75% prediction intervals
(Sources: [27,28]; authors' computation and illustration).

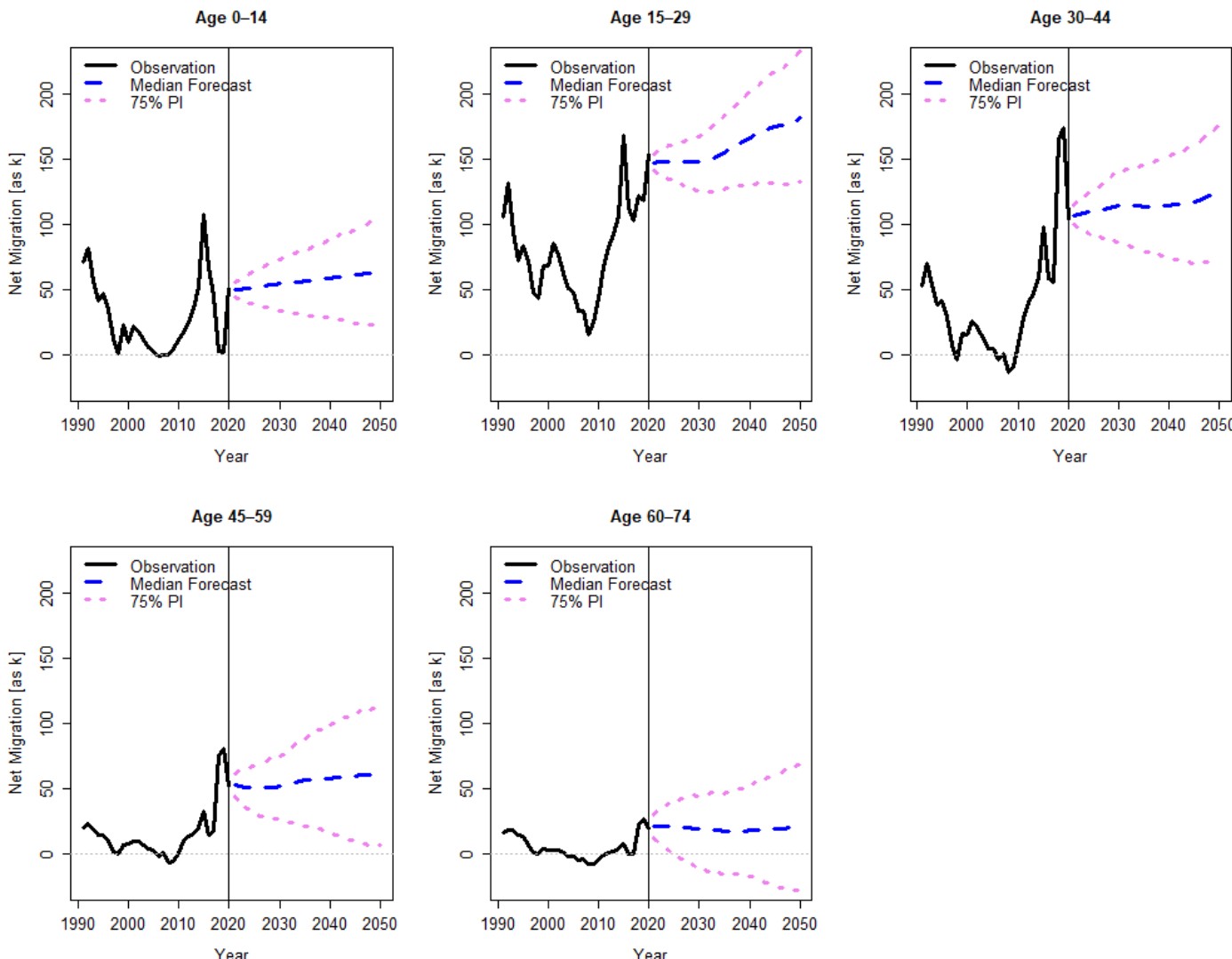

**Figure A4.** Forecasts of net migration of females by 15-year age groups with 75% prediction intervals (Sources: [27,28]; authors' computation and illustration).

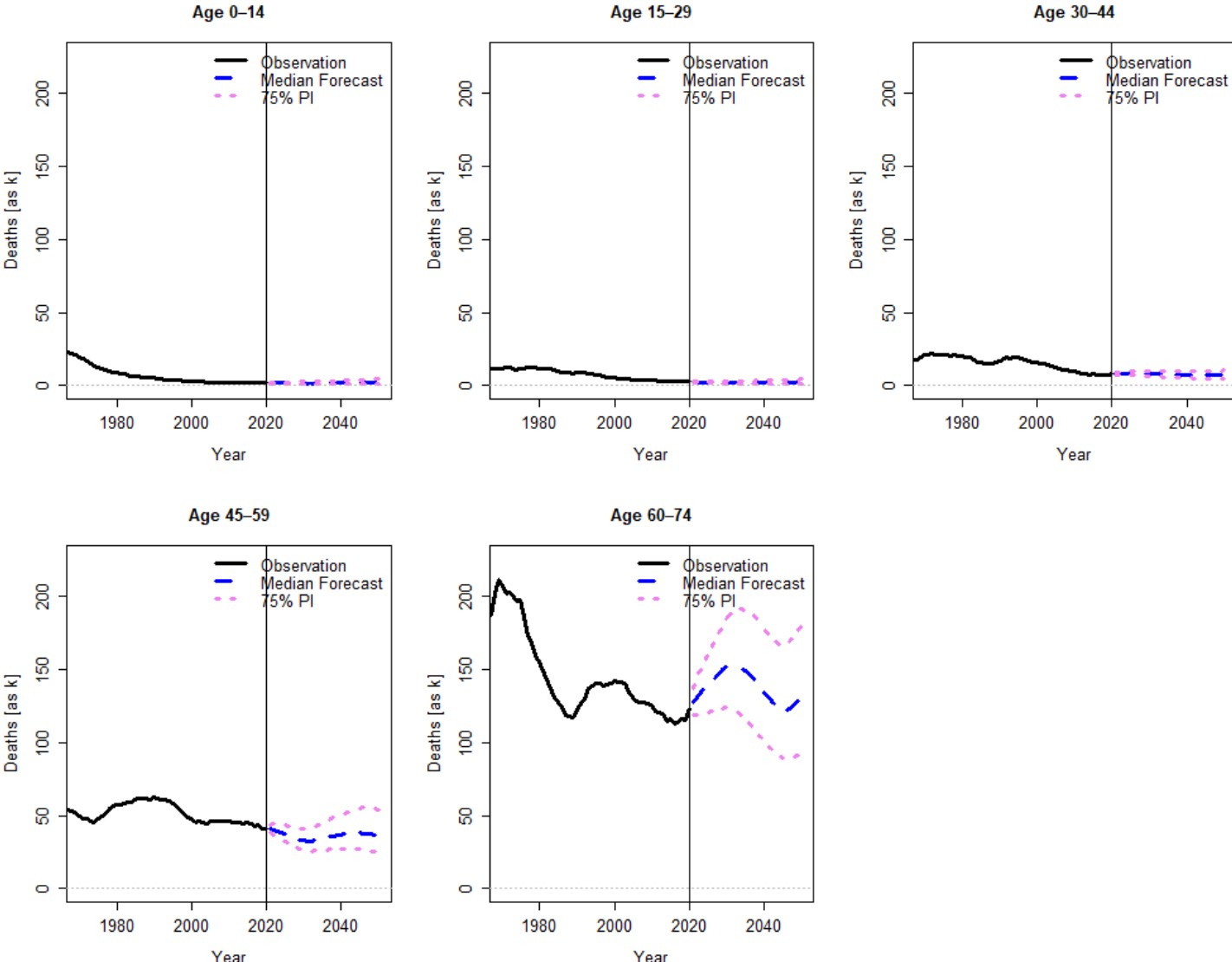

**Figure A5.** Forecasts of deaths of males by 15-year age groups with 75% prediction intervals (Sources: [29,30]; authors' computation and illustration).

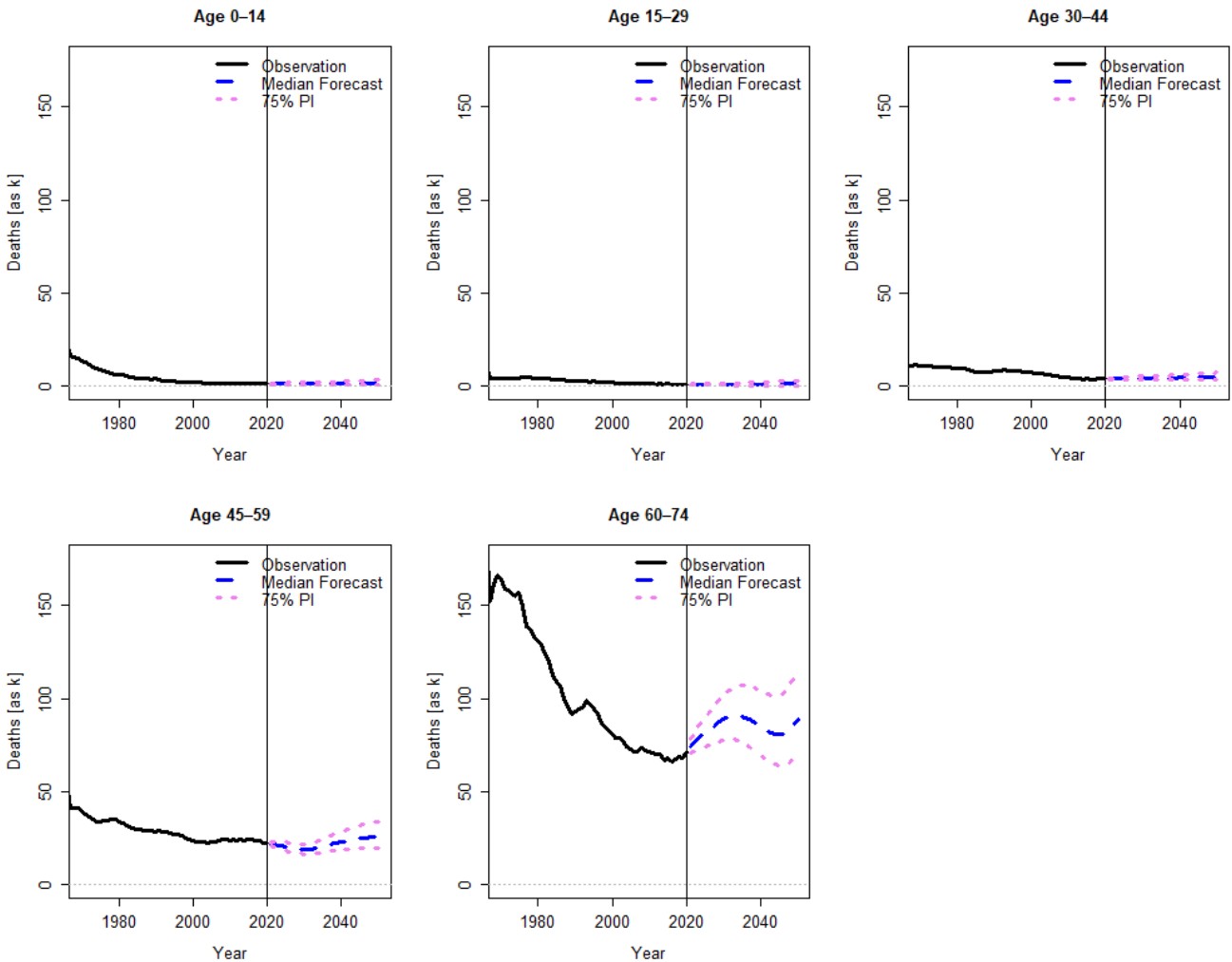

**Figure A6.** Forecasts of deaths of females by 15-year age groups with 75% prediction intervals (Sources: [29,30]; authors' computation and illustration).

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
