# Peer review of "Prevalence and Economic Costs of Absenteeism in an Aging Population—A Quasi-Stochastic Projection for Germany"

_forecasting, doi:10.3390/forecast4010021_

Round 1

Reviewer 1 Report

The paper examines the purely demographic effect on the economic costs associated with such absenteeism due to the inability to work. Under expected future employment patterns and constant morbidity patterns, absenteeism is expected by close to 5 percent by 2050 relative to 2020, associated with increasing economic costs of almost 3 percent.

Add more references from literature in the first section of the paper. Comparisons with other EU and non-EU countries are welcome.

Highlight better advantages and limitations of methodology.

Indicate alternative methods and explain why your method is better than others here.

In the discussion section indicate more comparisons with previous studies and highlight your value added.

Make a separate section for conclusions. Indicate novelty of your study, limitations and future directions of research.

Reviewer 2 Report

The paper investigates the important topic of absenteeism in Germany and its economic costs. While the topic has clearly some merit, I find that the statistical methodology of estimation is not well explained and the contribution seems to be a small addition to previous works of the same authors. More specifically:

  • The statistical method to estimate fertility and mortality is not explained. The authors refer to other papers but this paper should be self-contained and explained clearly how to reproduce all the results.
  • The authors often refer to Vanella and Deschermeier, and Vanella et al. It is not clear what the contribution is compared to these other papers.
  • In my opinion, the labor force participation rates are not sufficiently explained and illustrated. It would be beneficial to provide some illustrating graphs.
  • The Projection of Relative Increase in Absenteeism is only based on data of 2019, hence deterministic. It is an important limitation. I believe it is possible to extract data from several years, e.g. 2015-2019 and determine a trend by age group.

Reviewer 3 Report

Dear Authors, 

thank you for your interesting manuscript. I have some comments:

first, please correct the order of references according to the citation in the text.

second, because the thematic is very actual nowadays, could you add more explanation, your own interpretation. Your main text ended with the table (before the discussion part), maybe you could add the comment.

Round 2

Reviewer 2 Report

I am satisfied with the revised manuscript and thank the authors for the work.

Author Response

Dear Reviewer,

many thanks again for your careful reading of our paper and the helpful comments you made earlier.

We are very pleased that you found our revisions appropriate.

Best regards!

The Authors